# Targeting necroptosis protects against astrocyte death and hippocampal sclerosis in experimental temporal lobe epilepsy

Zhou Wu, Lukas Henning, Christian Steinhäuser and Peter Bedner

*Institute of Cellular Neurosciences I, Medical Faculty, University of Bonn, Bonn, Germany*

Handling Editors: Nathan Schoppa & Valentina Mosienko

The peer review history is available in the Supporting Information section of this article (https://doi.org/10.1113/JP287565#support-information-section).

**Abstract figure legend** Experimentally induced status epilepticus induces microglial TNFα release, which binds to astrocytic TNFR1 and triggers activation of RIPK1/RIPK3/MLKL signalling, initiating necroptotic astrocytic death that contributes to the development of hippocampal sclerosis. Created with Biorender.com.

**Abstract** Temporal lobe epilepsy (TLE) is the most common and severe form of adult focal epilepsy and is frequently associated with hippocampal sclerosis (HS). Accumulating evidence suggests that brain inflammation plays an important pathophysiological role in TLE. A key pro-inflammatory mediator is tumour necrosis factor $\alpha$ (TNF$\alpha$), which can trigger necroptosis pathways regulated by RIPK1, RIPK3 and MLKL through binding to the TNF receptor 1 (TNFR1). Previously, we detected activation of RIPK1, RIPK3 and MLKL in hippocampal CA1 astrocytes, along with a reduction in astrocyte density, during the early stages of experimentally induced TLE-HS, providing strong evidence for necroptotic astrocytic cell death. Using immunohistochemistry, pharmacology and long-term EEG recording, here we unravel the mechanisms underlying necroptosis induction in astrocytes and its role in epileptogenesis. The results show that pharmacological inhibition of necroptosis using Nec-1s, a specific inhibitor of RIPK1, or selective inhibition of soluble TNF$\alpha$ by XPro1595, as well as genetic knockout of TNFR1, effectively rescued CA1 astrocyte loss caused by kainate-induced status epilepticus. Furthermore, targeting necroptosis by Nec-1s administration attenuated CA1 astrogliosis, degeneration of pyramidal neurons, granular cell dispersion and shrinkage of the CA1 subfield. In contrast, Nec-1s did not affect acute and chronic epileptic activity in the TLE-HS model. Our findings demonstrate that TNF$\alpha$-induced necroptotic astrocyte death is involved in the pathogenesis of TLE.

(Received 30 January 2025; accepted after revision 19 June 2025; first published online 7 July 2025)

**Corresponding authors** C. Steinhäuser and P. Bedner: Institute of Cellular Neurosciences I, Medical Faculty, University of Bonn, Venusberg-Campus 1, 53127 Bonn, Germany. Email: Christian.steinhaeuser@uni-bonn.de, peter.bedner@ukbonn.de

## Key points

- In the early stage of experimental temporal lobe epilepsy with hippocampal sclerosis (TLE-HS), we observed activation of RIPK1, RIPK3 and MLKL in CA1 astrocytes, along with a reduction in astrocyte density, providing evidence for necroptotic cell death.
- Pharmacological inhibition of necroptosis (Nec-1s), blockade of soluble tumour necrosis factor $\alpha$ (XPro1595) or genetic knockout of TNFR1 prevented astrocyte loss after status epilepticus.
- Continuous telemetric EEG recordings showed that Nec-1s has no effect on acute or chronic epileptic activity in the TLE-HS model.
- Immunohistochemical analysis revealed that Nec-1s treatment reduced the extent of hippocampal sclerosis in experimental TLE-HS.
- Our results provide further insights into the molecular mechanisms underlying the development and progression of TLE.

## Introduction

Epilepsy affects 1–2% of the population worldwide (Hesdorffer et al., 2011). Temporal lobe epilepsy (TLE), the most prevalent and treatment-resistant form of focal epilepsy in adults, is often associated with severe atrophy of the hippocampus, a pathology known as hippocampal sclerosis (HS) (Blümcke et al., 1999). A major feature

**Zhou Wu** earned his Doctor of Medicine at the University of Bonn. During his doctoral studies and subsequent research experiences in the laboratory coordinated by Professor Christian Steinhäuser, he investigated the role of necroptosis in astrocytes in the context of the initiation and progression of epilepsy. Their findings strongly suggest that TNF$\alpha$-induced necroptotic death of astrocytes is involved in the pathogenesis of temporal lobe epilepsy. Subsequent to June 2023, he returned to China to continue his neuroscience research, now focusing on the consequences of coffee consumption and cognitive behavioural economics.

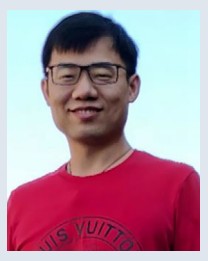

of HS is astrogliosis, a reactive process characterized by profound morphological, molecular and functional astrocytic changes (Blümcke et al., 2007; Escartin et al., 2021). Dysfunction of the astroglial network plays a critical role in the pathogenesis of TLE (Bedner et al., 2015; Deshpande et al., 2020). Astrocytes are electrically and metabolically connected to each other via gap junctions (Bedner & Steinhäuser, 2023). Through this coupling, they form large functional networks that are crucial for proper ion homeostasis, transmitter removal from the extracellular space and delivery of energy metabolites to neurons (Boison & Steinhäuser, 2018). Any reduction of astrocytic coupling has been suggested to induce or aggravate epilepsy (Bedner et al., 2015; Deshpande et al., 2020).

Our previous data demonstrated that in the unilateral intracortical kainate (KA) model of TLE, disruption of gap junction coupling and consequent impairment of $K^+$ homeostasis starts within a few hours after status epilepticus (SE) (Bedner et al., 2015). We found indications that proinflammatory cytokines, including tumour necrosis factor $\alpha$ (TNF$\alpha$), play a crucial role in this process (Bedner et al., 2015; Henning et al., 2023). TNF$\alpha$, mainly produced by reactive microglia after SE induction, specifically binds to TNF receptor 1 (TNFR1), which is upregulated in astrocytes 4 h after KA injection (Henning et al., 2023). The binding of TNF$\alpha$ to TNFR1 subsequently triggers activation of receptor-interacting protein kinase 1 (RIPK1) and a downstream signalling cascade inducing necroptosis (Yuan et al., 2019). Our recent research demonstrated that a considerable number of hippocampal CA1 astrocytes rapidly undergo necroptosis, followed by a repopulation within 3 days of KA injection (Wu et al., 2021). However, whether astrocytic necroptosis is involved in epileptogenesis is still unclear. The present study therefore aimed to investigate the molecular mechanism(s) triggering astrocyte necroptosis and to examine the role of hippocampal astrocytic death in the initiation and progression of experimental TLE.

## Methods

### Animals and ethical approval

Male C57B6/J (Charles River, Sulzfeld, Germany) and TNFR1 global knock-out (Jackson Laboratory, Bar Harbor, USA, #003242) mice (gKO) aged 90–120 days were used for the experiments. Mice were kept under standard housing conditions (12/12 h dark–light cycle, food and water *ad libitum*). Maintenance and handling of animals were performed according to EU and local governmental regulations. All experiments have been approved by the state of North Rhine Westphalia (Landesamt für Natur, Umwelt und Verbraucherschutz Nordrhein-Westfalen, approval numbers 84-02.04.2015.

A393 and 81-02.04.2020.A420). All measures were taken to minimize animal pain and suffering. The authors confirm their full understanding of *The Journal of Physiology*'s ethical principles and have complied with the animal ethical checklists.

### TLE model and EEG recordings

We used the TLE animal model as previously established (Bedner et al., 2015). Briefly, mice were anaesthetized with a mixture of medetomidine (Cepetor, CP-Pharma, Burgdorf, Germany, 0.3 mg/kg, I.P.) and ketamine (Ketamidor, WDT, Garbsen, Germany, 40 mg/kg I.P.) and placed in a stereotaxic frame equipped with a manual microinjection unit (David Kopf, Tujunga, CA, USA). Seventy nanolitres of a 20 mM solution of KA (Tocris, Bristol, UK) in 0.9% sterile NaCl was stereotactically injected into the neocortex above the right dorsal hippocampus. The stereotactic coordinates were 2 mm posterior to bregma, 1.5 mm from midline and 1.7 mm from the skull surface. Electrographic seizures were detected via skull surface electrodes implanted immediately after KA injection. For this, two holes were drilled 1 mm posterior to the injection site and 1.5 mm to the side of the midline to insert two monopolar leads for electrographic recording of seizures. Telemetric transmitters [TA10EA-F20 or TA11ETA-F10; Data Sciences International (DSI), St. Paul, MN, USA] were implanted subcutaneously in the right abdominal region, and the two monopolar leads were inserted ~1 mm deep into the cortex. The attached leads were secured to the skull with superglue and covered with dental cement. The scalp incision was then sutured, and anaesthesia was stopped with atipamezole (Antisedan, Orion Pharma, Hamburg, Germany, 300 mg/kg, I.P.). Mice were injected with carprofen (Rimadyl, Pfizer, Karlsruhe, Germany) for 3 days for pain relief. In addition, 0.25% enrofloxacin (Baytril, Bayer, Leverkusen, Germany) was administered through drinking water to reduce the risk of infection. After surgery, mice were returned to clean cages and placed on individual radio receiver plates (RPC-1; Data Sciences International, New Brighton, MN, USA), which captured data signals from the transmitter and transferred them to a computer running Ponemah software (version 5.2, DSI). EEG recordings (24 h/day, 7 days/week) were started immediately after transmitter implantation and continued for 28 days after SE induction.

### EEG data analysis

NeuroScore 3.4.0 software (DSI) was used for EEG analysis, as described previously (Deshpande et al., 2020; Henning et al., 2023). For quantification of SE, recordings during the first 4 h after KA were high-pass filtered at

1 Hz, and the number of EEG spikes exceeding 10-fold the SD of baseline activity (i.e. activity during artefact- and epileptiform-free epochs 4 weeks after SE) was assessed using the spike train analysis tool implemented in Neuro-Score. For spectral analysis, EEG data were processed using Fast Fourier Transformation (FFT), and the power of high-frequency activity ($\gamma$-band power: 30–50 Hz) was calculated for each second of the 4 h recording and subsequently normalized to baseline. To quantify spike activity during the latent and chronic phase, the EEG data from the entire 4 week period (excluding the first 4 h) were high-pass filtered at 1 Hz, and the number of spikes exceeding 7.5 times baseline activity (same baseline used for SE quantification) was counted. In addition, the EEG power of all frequency bands (0.5–50 Hz) was calculated in 10 s epochs after FFT and normalized to baseline. The number of spontaneous generalized seizures (SGS) during the chronic phase was determined manually from the EEG recordings by two experienced experimenters.

## Nec-1s and Xpro1595 *in vivo* treatment

To pharmacologically block astrocytic necroptosis *in vivo*, we used Nec-1s (specific RIPK1 inhibitor) and 1595 (specific soluble TNF$\alpha$ inhibitor). As previously reported (Caccamo et al., 2017; Ofengeim et al., 2015), Nec-1s (10 mg/kg) was administered to C57B6/J mice twice intraperitoneally (I.P.) at 2 h intervals immediately after KA injection. Control mice received vehicle injections under the same conditions. XPro1595 (Xencor, Monrovia, CA, USA) was injected I.P. three times (10 mg/kg) at 3 day intervals starting 7 days before KA injection.

## Tissue preparation and immunohistochemistry

Mice were deeply anaesthetized by I.P. injection of xylazine/ketamine and perfused intracardially with ice-cold phosphate-buffered saline (PBS) followed by 4% paraformaldehyde (PFA). Brains were dissected, post-fixed overnight in 4% PFA at 4°C, and transferred into PBS. Prefixed mouse brains were cut with a vibratome into 40 μm thick sections. After permeabilization and blocking [2 h at room temperature with 0.5% Triton X-100 and 10% normal goat serum (NGS) or 10% normal donkey serum (NDS) in PBS], the sections were incubated overnight (4°C) in 5% NGS (or NDS) in PBS containing 0.1% Triton X-100 and the following primary antibodies: mouse anti-GFAP (Merck Millipore, Darmstadt, Germany, 1:500, Cat.# MAB360, RRID: AB_11212597); goat anti-GFAP (Abcam, Berlin, Germany, 1:500, Cat.# ab302644, RRID: AB_3669042); goat anti-Iba1 (Abcam, 1:300, Cat.# ab5076, RRID: AB_2224402); and mouse anti-NeuN (Merck Millipore, 1:500, Cat.# MAB377, RRID: AB_2298772). After washing with PBS (5 min), the sections were incubated with secondary antibodies conjugated with Alexa Fluor 488, Alexa Fluor 555, Alexa Fluor 594 or Alexa Fluor 647 (Invitrogen, Karlsruhe, Germany, dilution 1:500 each) in PBS with 2.5% NGS (or 2.5% NDS) and 0.1% Triton X-100 (1.5 h, room temperature). After several washes in PBS, nuclei were stained with Hoechst (Sigma-Aldrich, Steinheim, Germany, 1:100 in distilled water).

## Quantification of immunostaining

**Image acquisition and cell counting.** Images were acquired at a resolution of $x = 1024$, $y = 1024$ pixels, using a confocal laser scanning microscope (Leica TCS SP8, Wetzlar, Germany). Using $63\times$ lenses, 30 sequential optical sections at 1 μm thickness were quantified. For each section, three representative adjacent and non-overlapping regions of interest (ROIs) were captured ($184.5 \times 184.5$ μm$^2$ for $63\times$ lenses; total ROI area $= 34,040$ μm$^2$) and digitized. ROIs were located in the CA1 stratum radiatum (s.r.), positioned side by side, as shown in a previous study (Wu et al., 2021). With $40\times$ lenses, sequential optical sections at 1 μm thickness were quantified. Cell counting in the dentate gyrus (DG) and CA1 regions was performed in $290.91 \times 290.91$ μm$^2$ counting boxes (total ROI area $= 84,628$ μm$^2$). Data obtained from the ROIs were averaged, providing a single value for each slice. Values obtained from three slices were averaged per mouse, and these data were used for statistical analysis. To avoid bias, analyses were performed independently by two colleagues blinded to the experimental conditions.

## Hippocampal sclerosis

As described previously (Deshpande et al., 2020) the severity of HS was measured at four different dimensions: (1) gliosis in the CA1 region, (2) number of surviving CA1 pyramidal neurons, (3) granule cell dispersion (GCD) in the DG and (3) shrinkage of the CA1 s.r. Briefly, the extent of gliosis was measured by quantifying the area occupied by GFAP and Iba1 immunoreactivity (IR) in individual ROIs within all focal planes of each image in CA1. The number of pyramidal neurons was measured within ROIs of $100 \times 290.91 \times 15$ μm$^3$ positioned in the CA1 pyramidal layer above the tip of the granule cell layer. GCD quantification was performed as described previously (Deshpande et al., 2020; Fig. 4*A*). The GCL width at four positions (T1–T4) was measured and averaged to determine GCD. Shrinkage of the CA1 s.r. was determined by drawing a vertical line connecting the pyramidal and molecular layer above the tip of the GCL. The length of the vertical line served as an indication of the remaining

width of the s.r. All parameters were quantified using Fiji software.

## Statistical analysis

Statistical analyses were conducted with SPSS 26.0 or OriginPro (OriginLab Corp., Northampton, MA, USA). Data are displayed as mean $\pm$ standard deviation (SD) or as box plots representing median (line) and quartiles (25th and 75th percentile) with whiskers extending to the highest and lowest values within 1.5 times the interquartile range (IQR). The Shapiro–Wilk test assessed data for normality. For comparing two independent groups, either Student's $t$ test or the Mann–Whitney $U$ test was applied. Comparisons involving more than two groups were analysed through one-way analysis of variance (ANOVA) followed by the *post hoc* Tukey test. For multifactorial data, two-way ANOVA followed by the *post hoc* Tukey test was carried out. The significance level was set at $P < 0.05$.

## Results

### Targeting necroptosis pathways prevents loss of hippocampal CA1 astrocytes in experimental TLE

In our previous work, we found necroptotic astrocytic cell death in the hippocampus of epileptic mice 4 h after SE induction through unilateral intracortical KA injection, with strong activation of RIPK3 and pMLKL (Wu et al., 2021). To unravel potential underlying mechanisms, we first tested whether pharmacological inhibition of RIPK1 by Nec-1s rescued SE-induced astrocyte loss. For this, Nec-1s was administered I.P. at a dose of 10 mg/kg directly and 2 h after KA, and the densities of Hoechst-labelled nuclei and GFAP-positive astrocytes in the ipsi- and contralateral CA1 region were quantified 4 h after SE induction (Fig. 1*A*). In agreement with our previous data (Wu et al., 2021), vehicle-treated mice showed ipsilateral loss of GFAP-positive (ipsi, 10,626.49 $\pm$ 965.03 mm$^{-3}$ *vs.* contra, 15,196.25 $\pm$ 844.66 mm$^{-3}$, $P < 0.001$, one-way ANOVA) and Hoechst-positive cells (ipsi, 36,866.32 $\pm$ 2565.02 mm$^{-3}$ *vs.* contra, 44,355.64 $\pm$ 3612.98 mm$^{-3}$, $P = 0.0015$, one-way ANOVA), 4 h after KA injection (Fig. 1*B*, *C* and *F*). Intriguingly, Nec-1s treatment prevented loss of both astrocytes (ipsi, 13,346.59 $\pm$ 1396.20 mm$^{-3}$ *vs.* contra, 14,198.88 $\pm$ 1017.77 mm$^{-3}$, $P = 0.53$, one-way ANOVA) and nuclei (ipsi, 40,638.18 $\pm$ 1912.90 mm$^{-3}$ *vs.* contra, 43,140.67 $\pm$ 3036.97 mm$^{-3}$, $P = 0.44$, one-way ANOVA) (Fig. 1*B*, *C* and *F*). Importantly, the number of GFAP-negative/Hoechst-positive cells did not differ under the different conditions (vehicle, ipsi, 26,203.5 $\pm$ 1986.6 *vs.* contra, 29,086.9 $\pm$ 3219.3, $P = 0.2$; Nec-1s, ipsi, 27,291.6 $\pm$ 1663.5 *vs.* contra,

28,947.8 $\pm$ 2647.1, $P = 0.65$, one-way ANOVA), indicating that only astrocytes died at this time point. Thus, hippocampal astrocytes undergo RIPK1-dependent necroptotic death in the early phase of experimental TLE.

### Activation of TNFR1 by soluble TNF$\alpha$ induces necroptotic astrocyte death

The TNF pathway is an important initiator of necroptosis and a number of studies have reported increased levels of TNF$\alpha$ in both human and experimental epilepsy (Benson et al., 2015; De Simoni et al., 2000; Eberhard et al., 2024; Henning et al., 2023; Lachos et al., 2011; Sano et al., 2021; Vezzani et al., 2002; Yamamoto et al., 2006). To test whether TNF$\alpha$ has a role in the induction of astrocytic necroptosis in our TLE model, TNF$\alpha$/TNFR1 signalling was inhibited by applying the dominant-negative soluble TNF$\alpha$ (sTNF$\alpha$) inhibitor XPro1595. This inhibitor is known to penetrate the blood–brain barrier after I.P. injection and preferentially blocks TNFR1 activation (Barnum et al., 2014; Steed et al., 2003). XPro1595-mediated depletion of soluble TNF$\alpha$ (evoked by three I.P. injections of 10 mg/kg every third day, starting 7 days prior to the injection of KA) prevented SE-induced loss of astrocytes (ipsi, 15,170.34 $\pm$ 1836.67 mm$^{-3}$ *vs.* contra, 15,481.21 $\pm$ 1471.49 mm$^{-3}$, $P = 0.73$, $t$ test) and Hoechst-stained nuclei (ipsi, 43,739.09 $\pm$ 2438.60 mm$^{-3}$ *vs.* contra, 45,153.53 $\pm$ 3325.79 mm$^{-3}$, $P = 0.38$, $t$ test) in the ipsilateral hippocampus, 4 h after KA (Fig. 1*D* and *F*). Moreover, no reduction in the number of GFAP-positive cells (ipsi 15,776.54 $\pm$ 1890.81 mm$^{-3}$ *vs.* contra 15,580.69 $\pm$ 1205.22 mm$^{-3}$, $P = 0.85$, $t$ test) or Hoechst-positive nuclei (ipsi 42,694.57 $\pm$ 2984.28 mm$^{-3}$ *vs.* contra 42,542.25 $\pm$ 1761.14 mm$^{-3}$, $P = 0.92$, $t$ test) was detectable in mice with global deletion of TNFR1 (Fig. 1*E* and *F*). Taken together, these data indicate that in experimental TLE, activation of TNFR1 by soluble TNF$\alpha$ activates RIPK1, causing necroptotic astrocyte loss in the hippocampal CA1 region.

### Inhibiting necroptosis does not affect acute or chronic epileptiform activity

To elucidate the role of necroptotic CA1 astrocytes in the initiation and progression of epilepsy, KA-injected mice were treated with vehicle or Nec-1s as described above and implanted with telemetric transmitters for continuous (24 h/day for 4 weeks) telemetric EEG recordings. Electrical activity during SE was quantified by counting the number of EEG spikes with amplitudes exceeding baseline activity at least 7.5-fold and by comparison of the spectral power in the $\gamma$ range after FFT of EEG data during the first 4 h of recording (Deshpande et al., 2020). We found no differences between KA-injected mice

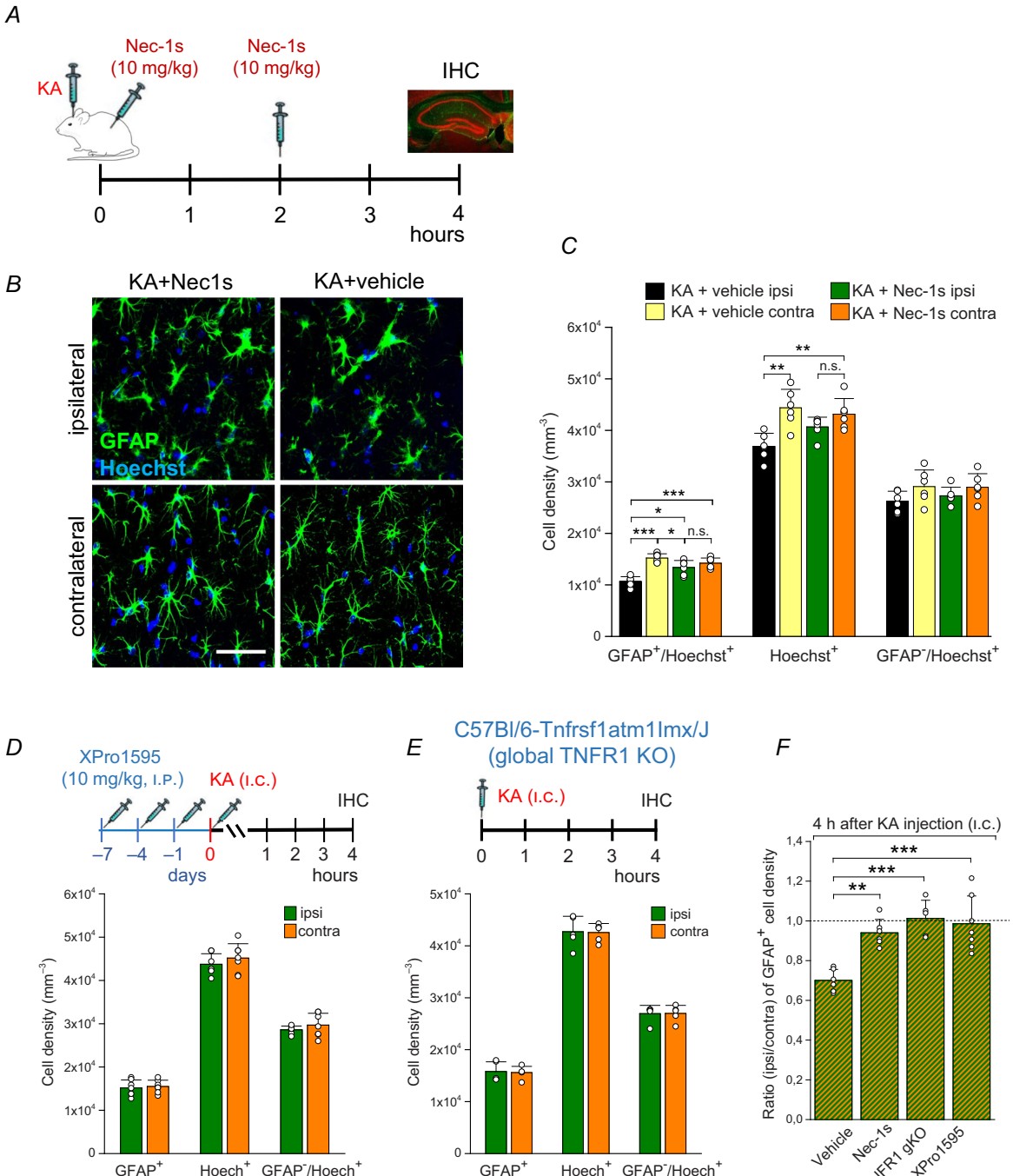

**Figure 1. Astrocytic loss in the CA1 stratum radiatum during early epileptogenesis can be prevented by targeting necroptosis and TNF/TNFR1 signalling**

*A*, schematic representation of the experimental procedure. TLE was induced in C57B6/J mice by stereotactic unilateral intracortical injection of KA, followed immediately by the first injection of the necroptosis inhibitor Nec-1s (10 mg/kg, I.P.). The second Nec-1s injection was given 2 h later, and immunohistochemical analyses were performed 4 h after KA injection. *B*, representative photomicrographs of GFAP (green) and Hoechst (blue) double staining in hippocampal slices from vehicle- and Nec-1s-treated mice 4 h after KA injection. Scale bar = 50 μm. *C*, quantification of the number of GFAP-positive/Hoechst-positive, Hoechst-positive and GFAP-negative/Hoechst-positive cells per region of interest (184.5 × 184.5 × 30 μm$^3$) within the CA1 stratum radiatum in vehicle- and Nec-1s-treated animals. *N* = 6 animals per condition. *D*, the sTNFα inhibitor XPro1595 was injected every 72 h (10 mg/kg, I.P.) starting 7 days before KA-mediated induction of TLE. No reduction in GFAP- or Hoechst-positive cells was found in the CA1 stratum radiatum. Four hours after KA injection in XPro1595-treated

mice ($N = 7$) or (*E*) in mice with global TNFR1 deletion ($N = 5$). *F*, quantification of the ratio (ipsi- over contralateral) of the GFAP-positive cell density within ROIs in the CA1 region [ratios, vehicle: $0.7 \pm 0.056$; Nec-1s: $0.94 \pm 0.068$; TNFR1 gKO: $1.01 \pm 0.09$; XPro1595: $0.98 \pm 0.14$; $P = 0.002$ (vehicle *vs.* Nec-1s); $P < 0.001$ (vehicle *vs.* TNFR1 gKO); $P < 0.001$ (vehicle *vs.* XPro15495)]. *$P < 0.05$, **$P < 0.01$, ***$P < 0.001$ (one-way ANOVA). Abbreviations: contra, contralateral; IHC, immunohistochemistry; ipsi, ipsilateral; n.s., not significant.

treated with Nec-1s or vehicle regarding the number of spikes (vehicle, $25.48 \pm 15.3$ spikes/min. *vs.* Nec-1s, $30.1 \pm 15.3$ spikes/min, $P = 0.58$, *t* test) or normalized $\gamma$ band power (vehicle, $5.6 \pm 2.4$ *vs.* Nec-1s, $4.4 \pm 2.3$, $P = 0.38$, *t* test) (Fig. 2*A* and *B*). Four out of seven (57.14 %) control mice and 5/7 (71.43%) mice treated with Nec-1 developed SGS (Fig. 2*C*). The total number of SGS during the 4 week recording period was highly variable between individual mice and did not differ significantly between conditions (vehicle, $9.4 \pm 14.3$ *vs.* Nec-1s, $9.57 \pm 10.47$, $P = 0.78$, Mann–Whitney *U* test; Fig. 2*D*, left panel). Likewise, neither the number of epileptic spikes (vehicle, $20.03 \pm 8.7$ spikes/min *vs.* Nec-1s, $21.1 \pm 8.5$ spikes/min, $P = 0.81$, *t* test; Fig. 2*D*, middle) nor normalized total power (vehicle, $2.2 \pm 0.3$ *vs.* Nec-1s, $2.3 \pm 0.87$, $P = 0.72$,

*t* test; Fig. 2*D*, right) differed throughout the recording period. These data indicate that the genesis of acute and chronic ictal as well as interictal activity is not affected by necroptotic cell death.

### Nec-1s attenuates HS development in KA-injected mice

TLE-HS is characterized by reactive astro- and microgliosis, neuronal loss in the CA1 region and GCD (Blümcke et al., 2007). These typical histopathological changes are well recapitulated in our TLE model (Bedner et al., 2015). To determine whether necroptosis affects these KA-induced morphological changes, we measured reactive astrogliosis and microgliosis in the CA1 region

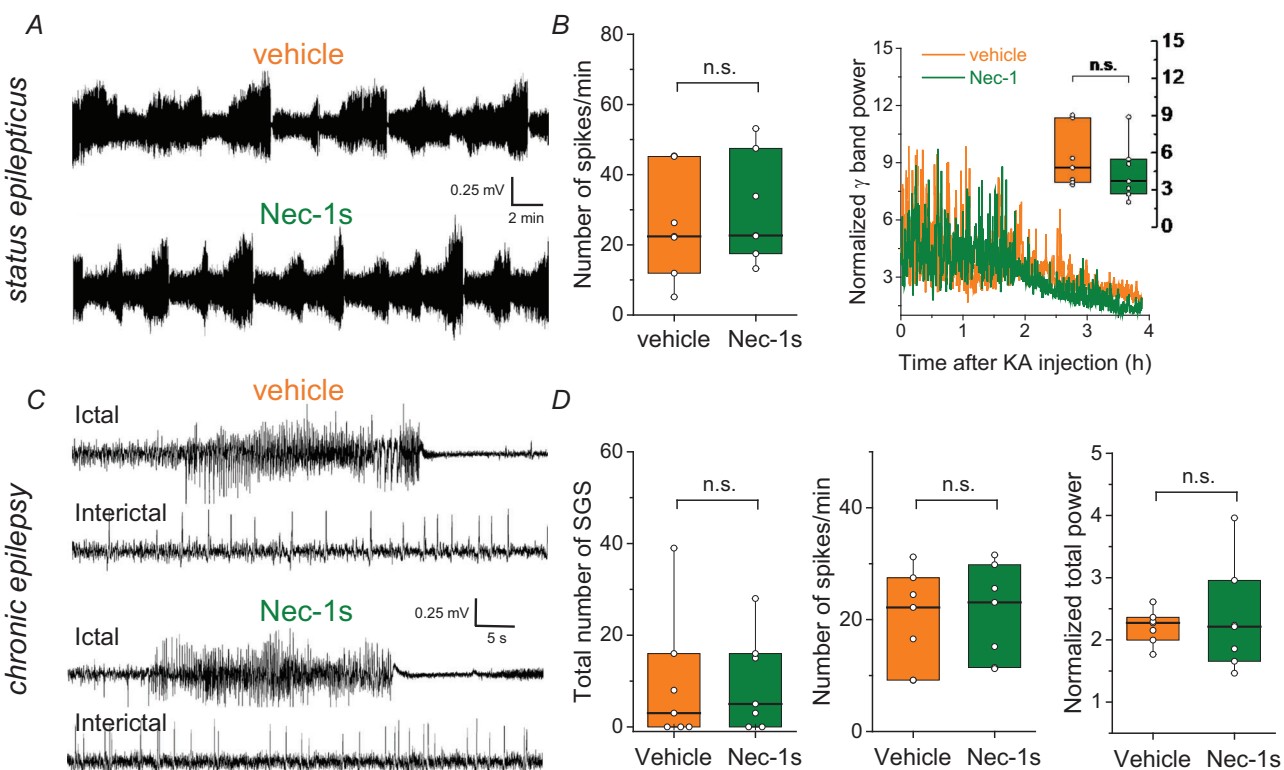

**Figure 2. Nec-1s does not affect epiletiform activity**
*A*, representative EEG traces recorded during SE (i.e. first 4 h after KA injection) in vehicle- and Nec-1s-treated KA-injected mice. *B*, severity of SE under the two conditions was quantified by spike frequency and spectral analysis of EEG data recorded during the first 4 h after KA injection. Neither spike activity nor $\gamma$ band power differed between vehicle- and Nec-1s-treated mice. *C*, representative EEG traces showing SGS (ictal) and interictal spiking activity (interictal) during the chronic phase of KA-induced epileptogenesis. Nec-1s treatment had no effect on SGS frequency, number of spikes or normalized total power during the 4 week recording period. $N = 7$ animals per condition. Abbreviation: n.s., not significant (Mann–Whitney *U* test and *t* test).

based on the area occupied by GFAP IR as reported previously (Deshpande et al., 2020). GFAP labelling was increased ipsi- *vs.* contralaterally in vehicle- but not in Nec-1s-treated mice (vehicle, ipsi 27.13 ± 4.10% *vs.* contra, 11.03 ± 6.60%, $P < 0.001$; Nec-1s: ipsi,

16.20 ± 5.93% *vs.* contra, 12.50 ± 3.40%, $P = 0.058$, two-way ANOVA) (Fig. 3*A* and *B*). Accordingly, the ratio (ipsi- over contralateral) of the area occupied by GFAP IR in the CA1 region was also increased in KA-injected vehicle- *vs.* Nec-1s-treated mice (vehicle: 2.94 ± 1.09

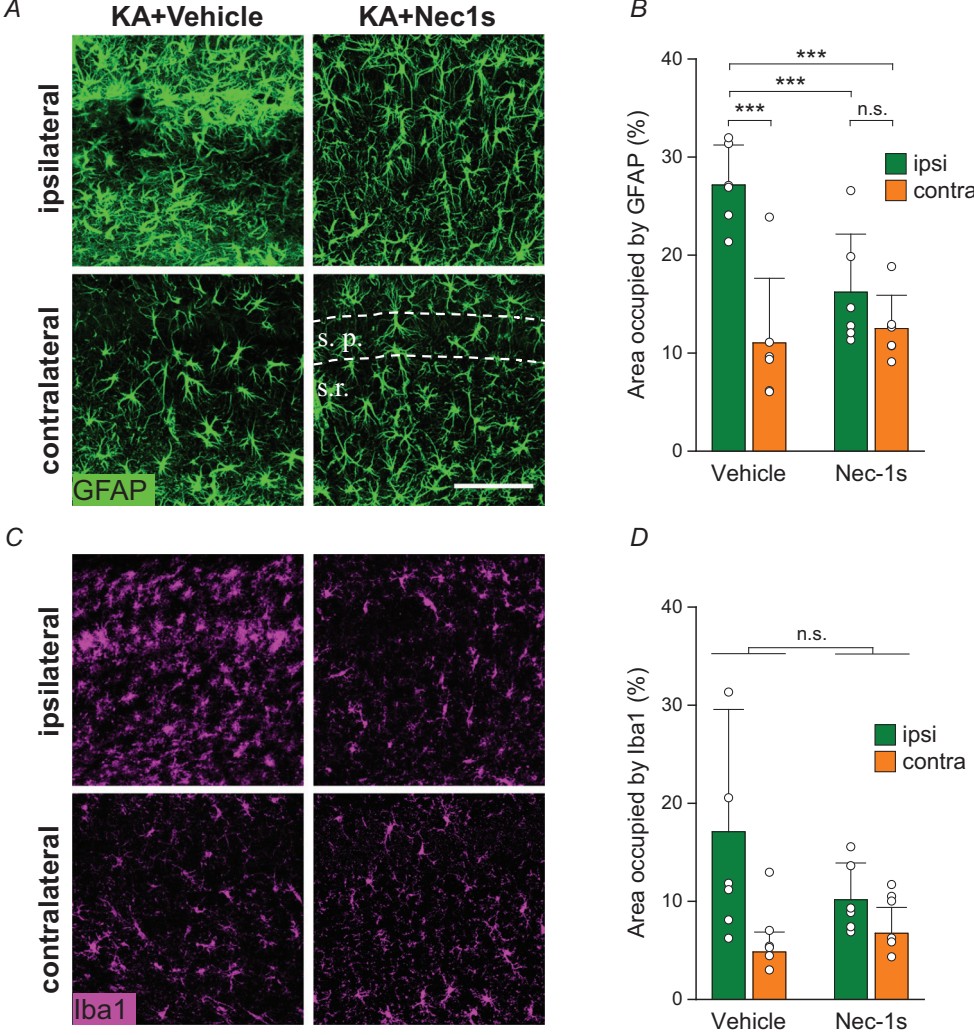

**Figure 3. Necroptosis promotes reactive astrogliosis in the CA1 region during the chronic phase of experimental TLE**

*A*, representative images of GFAP (green) staining in hippocampal slices from vehicle- and Nec-1s-treated mice, 4 weeks after KA injection. Scale bar = 100 μm. *B*, quantification of GFAP IR in ROIs of 290.91 × 290.91 × 15 μm³ within the CA1 subfield. Four weeks after KA injection, the extent of astrogliosis was significantly higher ipsi- *vs.* contralaterally in vehicle-treated mice, but not in mice treated with Nec-1s. *C*, representative images of Iba1 (magenta) staining in hippocampal slices from both conditions, 4 weeks after KA injection. *D*, quantification of Iba1 IR in ROIs of 290.91 × 290.91 × 15 μm³ within the CA1 region revealed no difference between vehicle- and Nec-1s-treated mice. ***$P < 0.001$, two-way ANOVA and Tukey test. $N = 6$ animals in each group. Abbreviations: contra, contralateral side; ipsi, ipsilateral side; s.p., stratum pyramidale; s.r., stratum radiatum.*A*, representative images of GFAP (green) staining in hippocampal slices from vehicle- and Nec-1s-treated mice, 4 weeks after KA injection. Scale bar = 100 μm. *B*, quantification of GFAP IR in ROIs of 290.91 × 290.91 × 15 μm³ within the CA1 subfield. Four weeks after KA injection, the extent of astrogliosis was significantly higher ipsi- *vs.* contralaterally in vehicle-treated mice, but not in mice treated with Nec-1s. *C*, representative images of Iba1 (magenta) staining in hippocampal slices from both conditions, 4 weeks after KA injection. *D*, quantification of Iba1 IR in ROIs of 290.91 × 290.91 × 15 μm³ within the CA1 region revealed no difference between vehicle- and Nec-1s-treated mice. ***$P < 0.001$, two-way ANOVA and Tukey test. $N = 6$ animals in each group. Abbreviations: contra, contralateral side; ipsi, ipsilateral side; s.p., stratum pyramidale; s.r., stratum radiatum.

*vs.* Nec-1s: $1.30 \pm 0.28$, $P = 0.013$, *t* test). Microglia reactivity detected by Iba1 IR was not affected by the Nec-1s treatment [vehicle, ipsi $17.1 \pm 12.47\%$ *vs.* contra, $4.83 \pm 2.03\%$; Nec-1s: ipsi, $10.15 \pm 3.75\%$ *vs.* contra, $6.74 \pm 2.64\%$, $P = 0.53$ (vehicle *vs.* Nec-1s), two-way ANOVA] (Fig. 3*C* and *D*).

In the next step, we determined the extent of neuronal loss in the CA1 region of vehicle- and Nec-1s-treated mice by NeuN and Hoechst staining, 4 weeks after KA injection (Fig. 4*A* and *B*). Vehicle-treated mice showed pronounced ipsilateral neurodegeneration (density of CA1 pyramidal neurons: ipsi, $9828.86 \pm 8636.71$ mm$^{-3}$ *vs.* contra, $25\,431.32 \pm 1425.2$ mm$^{-3}$, $P = 0.009$), which was less pronounced in Nec-1s-treated mice (ipsi, $21{,}798.7 \pm 3284.53$ mm$^{-3}$ *vs.* contra, $25{,}593.33 \pm 859.10$ mm$^{-3}$, $P = 0.018$; vehicle ipsi *vs.* Nec-1s ipsi, $P = 0.009$, two-way ANOVA) (Fig. 4*B* and *C*). Remarkably, GCD and shrinkage of the s.r. were completely prevented by Nec-1s treatment (width of GCL in vehicle: ipsi, $161.46 \pm 79.26$ µm *vs.* contra, $67.71 \pm 16.38$ µm, $P = 0.033$; Nec-1s: ipsi, $62.96 \pm 17.61$ µm *vs.* contra, $71.89 \pm 13.65$ µm; $P = 0.68$; width of s.r. in vehicle: ipsi, $134.85 \pm 47.44$ µm *vs.* contra, $190.12 \pm 20.05$ µm, $P = 0.028$; Nec-1s: ipsi, $189.20 \pm 22.84$ µm *vs.* contra, $186.56 \pm 24.07$ µm, $P = 0.83$, two-way ANOVA (Fig. 4*A* and *D–F*). Together, these data demonstrate that inhibition of necroptotic astrocytic cell death by Nec-1s attenuates the extent of HS in experimental TLE induced by intracortical KA injection.

## Discussion

The aim of our study was to investigate the mechanisms mediating SE-induced astrocyte loss and its contribution to the initiation and progression of experimental TLE. Here, we extend previous evidence (Wu et al., 2021) of activation of the necroptotic machinery in astrocytes of mice intracortically injected with KA. Notably, systemic injection of the specific necroptosis inhibitor Nec-1s almost completely prevented the loss of hippocampal astrocytes. The remaining slight reduction in astrocyte density in mice treated with Nec-1s (Fig. 1*C*) might indicate either incomplete inhibition of necroptosis or the presence of an additional mechanism of astrocytic death. Indeed, our previous work showed that a small fraction of astrocytes in the ipsilateral hippocampus express autophagy-related proteins at this stage of KA-induced epileptogenesis (Wu et al., 2021), which could account for the residual reduction. Further studies are needed to determine whether this is actually the case, and if so, why some astrocytes die by necroptosis and others by autophagy, what signals trigger autophagy and what role autophagic astrocytic death plays in the genesis of TLE.

Our study clearly demonstrates that astrocytic cell death results from TNFR1 activation by soluble TNFα. TNF-α is initially produced as a transmembrane protein (tmTNFα), which is cleaved by the metalloprotease TACE to produce the soluble 17 kDa form (sTNFα). It is assumed that tmTNFα predominantly activates TNFR2, promoting cell survival and immune regulation, while sTNFα activates TNFR1, which can induce apoptosis or necroptosis through its death domain (MacEwan, 2002; Santello & Volterra, 2012). The balance between TNFR1 and TNFR2 signalling, along with the intensity and duration of TNFα concentration increase, plays a crucial role in shaping inflammatory responses and determines whether the cytokine exerts protective or harmful effects (Santello & Volterra, 2012). Numerous studies have highlighted the significance of TNFα signalling in the development and progression of epilepsy (Balosso et al., 2005; Eberhard et al., 2024; Kirkman et al., 2010; Patel et al., 2017; Weinberg et al., 2013). These reports consistently indicate that sTNFα and TNFR1 activation promote neuronal hyperactivity and neurodegeneration, while activation of TNFR2 shows protective effects in epilepsy (Balosso et al., 2005; Patel et al., 2017; Weinberg et al., 2013). Previous work from our group demonstrated strongly increased ipsilateral hippocampal TNFα levels 4 h after intracortical KA injection and revealed that reactive microglia are the primary source of TNFα, although microglia-specific knockout of the cytokine had no effect on epileptogenesis (Henning et al., 2023). The differences between our previous study and the results in Nec-1s-treated mice are probably due to the selective inhibition of necroptosis in the latter, which is a downstream effect of sTNFα/TNFR1 signalling, whereas microglia-specific TNFα knockout mice also lacked the protective effect of TNFR2. The present approach may be advantageous because inhibition of necroptosis leaves the TNFR2 pathway and its associated beneficial effects intact, which may explain the attenuated neuropathological findings observed in this study.

The mechanism by which TNFα/TNFR1 signalling promotes epilepsy is still unclear, but different possible scenarios have been described. In addition to the potential direct effects of elevated TNFα levels on neuronal activity, several TNFα-induced changes in astrocytes have been identified, including decreased glutamate uptake, increased gliotransmission and disruption of gap junctional coupling between astrocytes, all of which are essential for regulating neuronal activity and preventing hyperactivity and seizures (Habbas et al., 2015; Henning et al., 2023; Nikolic et al., 2018; Santello & Volterra, 2012). Here, we show that in addition to these changes, some astrocytes undergo cell death, leading to a reduction in the effectiveness of their crucial regulatory functions, which can result in neurodegeneration and hyperactivity. Astrocytes play a key role in the uptake

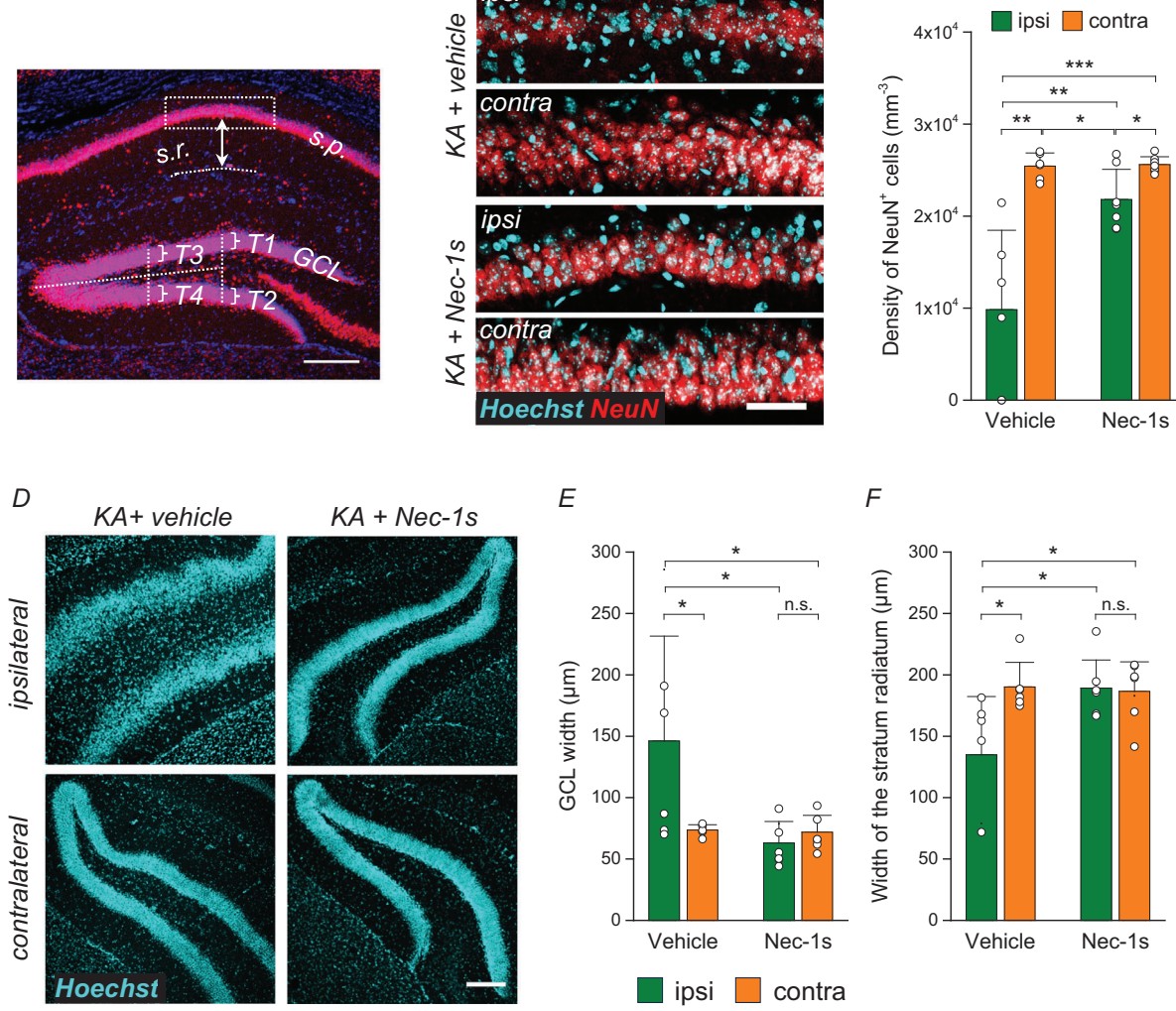

**Figure 4. Nec-1s treatment attenuates SE-induced hippocampal sclerosis**
*A*, illustration of the quantification method used to assess the extent of hippocampal sclerosis. *B*, representative photomicrographs of NeuN (red) and Hoechst (cyan) staining in the CA1 pyramidal cell layer of vehicle- and Nec-1s-treated mice, 4 weeks after KA injection. Scale bar = 50 μm. *C*, quantitative analysis of NeuN-positive cells in $100 \times 290.91 \times 15$ μm³ ROIs within the CA1 region revealed an ipsilateral decline in vehicle-treated mice 4 weeks after KA injection. In Nec-1s-treated mice, the extent of neurodegeneration was significantly lower. *D*, representative photomicrographs of Hoechst (cyan) staining in the DG of vehicle- and Nec-1s-treated mice, 4 weeks after KA injection. Scale bar = 200 μm. *E* and *F*, quantitative analysis of GCD and shrinkage of the CA1 stratum radiatum. In vehicle-treated mice, 4 weeks after KA injection, the ipsilateral hippocampus exhibited enlarged GCD and shrinkage compared to the contralateral side. Nec-1s treatment prevented both pathological changes. *$P < 0.05$, **$P < 0.01$ (two-way ANOVA and Tukey test. $N = 6$ (vehicle) and 7 (Nec-1s) mice. Abbreviations: contra, contralateral side; GCL, granule cell layer; ipsi, ipsilateral side. s.p., stratum pyramidale; s.r., stratum radiatum; T1–T4, the positions used to evaluate the width of the granule cell layer.*A*, illustration of the quantification method used to assess the extent of hippocampal sclerosis. *B*, representative photomicrographs of NeuN (cyan) staining in the CA1 pyramidal cell layer of vehicle- and Nec-1s-treated mice, 4 weeks after KA injection. Scale bar = 50 μm. *C*, quantitative analysis of NeuN-positive cells in $100 \times 290.91 \times 15$ μm³ ROIs within the CA1 region revealed an ipsilateral decline in vehicle-treated mice 4 weeks after KA injection. In Nec-1s-treated mice, the extent of neurodegeneration was significantly lower. *D*, representative photomicrographs of Hoechst (cyan) staining in the DG of vehicle- and Nec-1s-treated mice, 4 weeks after KA injection. Scale bar = 200 μm. *E* and *F*, quantitative analysis of GCD and shrinkage of the CA1 stratum radiatum. In vehicle-treated mice, 4 weeks after KA injection, the ipsilateral hippocampus exhibited enlarged GCD and shrinkage compared to the contralateral side. Nec-1s treatment prevented both pathological changes. *$P < 0.05$, **$P < 0.01$ (two-way ANOVA and Tukey test. $N = 6$ (vehicle) and 7 (Nec-1s) mice. Abbreviations: contra, contralateral side; GCL, granule cell layer; ipsi, ipsilateral side. s.p., stratum pyramidale; s.r., stratum radiatum; T1–T4, the positions used to evaluate the width of the granule cell layer.

and recycling of synaptically released glutamate, a critical process that ensures normal excitatory neurotransmission and protects against excitotoxicity (Boison & Steinhäuser, 2018; Schousboe, 2020). Other protective functions of astrocytes that are compromised by a reduced astrocyte density include the regulation of extracellular $K^+$ concentrations through spatial buffering and the control of adenosine levels via the enzyme adenosine kinase (Beamer et al., 2021; Bedner & Steinhäuser, 2023). In contrast, increased astrocytic release of gliotransmitters such as glutamate or ATP has also been discussed as a possible TNF$\alpha$-mediated epilepsy-promoting mechanism (Nikolic et al., 2018; Vezzani et al., 2022). A reduced astrocyte density would attenuate this pathological mechanism. Moreover, astrocytes are also capable of releasing the inhibitory neurotransmitter GABA (Müller et al., 2020), a function that has anti-epileptic effects and may be less efficient in case of reduced astrocyte densities.

Our previous work has shown that immediately after its early loss, astrocytes start to proliferate and recover within 3 days (Wu et al., 2021). It is therefore conceivable that the impact of astrocytic cell death is not solely due to the loss of cells, but rather to the emergence of new cells with altered, potentially epilepsy-promoting properties. In fact, we found less pronounced astrogliosis after Nec-1s treatment, suggesting that astrocytic cell death actually affects the properties of astrocytes in the long term.

It is interesting to note that, despite the reduced histopathological changes following treatment with Nec-1s, the inhibitor did not influence EEG activity. There are several possible explanations for this. First, we cannot exclude that the high variability in epileptic activity between mice hindered resolving putative Nec-1s effects. Second, the inhibitor was only administered twice at a very early stage, and since the half-life of the substance *in vivo* is very short ($\sim$1 h; Cao & Mu, 2021), astrocytic death may still have occurred with some delay. It is therefore possible that the role of astrocytic death in epileptogenesis was underestimated in our approach. Third, the surviving astrocytes in our model show impaired ion, glutamate and GABA homeostasis, and we do not know whether suppression of astrocytic necroptosis normalizes these homeostatic dysfunctions sufficiently to obtain a measurable effect on epileptiform activity. This question needs to be addressed in future work. Fourth, we did not analyse putative necroptotic astrocytic cell death in other brain regions. It cannot be completely excluded that astrocytic necroptosis also occurred in other associated brain regions, such as the amygdala and the entorhinal cortex, and that it was less effectively suppressed by Nec-1s in those regions. Finally, Nec-1s was applied systemically and may have affected necroptosis in cell types other than astrocytes. One approach to overcome this limitation would be to use astrocyte-specific and inducible knockout/down of RIPK1. This strategy would also allow us to study the effects of astrocyte-specific blockade of necroptosis on epileptogenesis.

In summary, our data show that at an early stage of experimental TLE, TNF/TNFR1 signalling triggers astrocytic necroptosis, which influences the process of epileptogenesis. Our results thus provide further insights into the molecular mechanisms of TLE development, which could be helpful for generating more specific and effective anti-epileptogenic drugs.

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

## Additional information

### Data availability statement

The data that support the findings of this study are available from the corresponding author upon reasonable request.

### Competing interests

None declared.

### Author contributions

Experiments were performed in the Institute of Cellular Neurosciences I, Medical Faculty, University of Bonn, Bonn, Germany. Z.W.: acquisition, analysis and interpretation of data, drafting the work; L.H.: acquisition, analysis and interpretation of data, drafting and revising the work; C.S.: conception and design, analysis and interpretation of data, drafting and revising the work; P.B.: conception and design, acquisition, analysis and interpretation of data, drafting and revising the work. All authors approved the final version of the manuscript and agree to be accountable for all aspects of the work in ensuring that questions related to the accuracy or integrity of any part of the work are appropriately investigated and resolved. All authors qualify for authorship, and all those who qualify for authorship are listed.

### Funding

This work was supported by Deutsche Forschungsgemeinschaft (grant STE 552/9 to C.S.).

### Acknowledgements

We thank INmune Bio, Boca Raton, FL, USA, for provision of XPro1595.

### Author's present address

L. Henning: Department of Epileptology, University Hospital Bonn, Bonn, Germany.

### Keywords

astrocyte, epileptogenesis, Nec-1s, necroptosis, temporal lobe epilepsy, TNF$\alpha$

## Supporting information

Additional supporting information can be found online in the Supporting Information section at the end of the HTML view of the article. Supporting information files available:

**Peer Review History**

