## [Peer Review History · The Journal of Physiology]

Targeting necroptosis protects against astrocyte death and hippocampal sclerosis in experimental temporal lobe epilepsy

Zhou Wu, Lukas Henning, Christian Steinhäuser, and Peter Bedner

DOI: 10.1113/JP287565

Corresponding author(s): Christian Steinhäuser (christian.steinhaeuser@uni-bonn.de)

The following individual(s) involved in review of this submission have agreed to reveal their identity: Bhanu Prakash Tewari (Referee #1)

Review Timeline:

Submission Date:	30-Jan-2025
Editorial Decision:	09-Apr-2025
Revision Received:	06-May-2025
Accepted:	19-Jun-2025

Senior Editor: Nathan Schoppa

Reviewing Editor: Valentina Mosienko

Transaction Report:

Dear Dr Steinhäuser,

Re: JP-RP-2025-287565 "Targeting necroptosis protects against astrocyte death and hippocampal sclerosis in experimental temporal lobe epilepsy" by Zhou Wu, Lukas Henning, Christian Steinhäuser, and Peter Bedner

Thank you for submitting your manuscript to The Journal of Physiology. It has been assessed by a Reviewing Editor and by 1 expert referee and we are pleased to tell you that it is potentially acceptable for publication following satisfactory major revision.

Please address all the points raised and incorporate all requested revisions or explain in your Response to Referees why a change has not been made. We hope you will find the comments helpful and that you will be able to return your revised manuscript within 2 months. If you require longer than this, please contact journal staff: jp@physoc.org. Please note that this letter does not constitute a guarantee for acceptance of your revised manuscript.

REVISION CHECKLIST:

Please upload two versions of your manuscript text: one with all relevant changes highlighted and one clean version with no

changes tracked. The manuscript file should include all tables and figure legends, but each figure/graph should be uploaded as separate, high-resolution files.

We look forward to receiving your revised submission.

Yours sincerely,

Nathan Schoppa
Senior Editor
The Journal of Physiology

REQUIRED ITEMS

- Author photo and profile. First or joint first authors are asked to provide a short biography (no more than 100 words for one author or 150 words in total for joint first authors) and a portrait photograph. These should be uploaded and clearly labelled together in a Word document with the revised version of the manuscript. See Information for Authors for further details.

- You must start the Methods section with a paragraph headed Ethical approval (https://jp.msubmit.net/cgi-bin/main.plex?form_type=display_requirements#methods).

Research must comply with The Journal's policies regarding animal experiments (<https://physoc.onlinelibrary.wiley.com/hub/animal-experiments>) and adherence to these policies must be stated in the manuscript.

Authors should confirm in their Methods section that their experiments were carried out according to the guidelines laid down by their institution's animal welfare committee, including an ethics approval reference number. The Methods section must contain a statement about access to food, water and housing, details of the anaesthetic regime: anaesthetic used, dose and route of administration, and method of killing the experimental animals.

- Your manuscript must include a complete Additional Information section, including competing interests; funding; author contributions and acknowledgements.

- Please upload separate high-quality figure files via the submission form.

- Please ensure that the Article File you upload is a Word file.

- Papers must comply with the Statistics Policy: https://jp.msubmit.net/cgi-bin/main.plex?form_type=display_requirements#statistics.

In summary:

- If $n \leq 30$, all data points must be plotted in the figure in a way that reveals their range and distribution. A bar graph with data points overlaid, a box and whisker plot or a violin plot (preferably with data points included) are acceptable formats.

- If $n > 30$, then the entire raw dataset must be made available either as supporting information, or hosted on a not-for-profit repository, e.g. FigShare, with access details provided in the manuscript.

- 'n' clearly defined (e.g. x cells from y slices in z animals) in the Methods. Authors should be mindful of pseudoreplication.
- All relevant 'n' values must be clearly stated in the main text, figures and tables.
- The most appropriate summary statistic (e.g. mean or median and standard deviation) must be used. Standard Error of the Mean (SEM) alone is not permitted.
- Exact p values must be stated. Authors must not use 'greater than' or 'less than'. Exact p values must be stated to three significant figures even when 'no statistical significance' is claimed.

- Please include an Abstract Figure file, as well as the Figure Legend text within the main article file. The Abstract Figure is a piece of artwork designed to give readers an immediate understanding of the research and should summarise the main conclusions. If possible, the image should be easily 'readable' from left to right or top to bottom. It should show the physiological relevance of the manuscript so readers can assess the importance and content of its findings. Abstract Figures should not merely recapitulate other figures in the manuscript. Please try to keep the diagram as simple as possible and without superfluous information that may distract from the main conclusion(s). Abstract Figures must be provided by authors no later than the revised manuscript stage and should be uploaded as a separate file during online submission labelled as File Type 'Abstract Figure'. Please also ensure that you include the figure legend in the main article file. All Abstract Figures should be created using BioRender. Authors should use The Journal's premium BioRender account to export high-resolution images. Details on how to use and access the premium account are included as part of this email.

EDITOR COMMENTS

Reviewing Editor:

Ethics Concerns:

Please make sure your article complies with our animal ethics policy: <https://physoc.onlinelibrary.wiley.com/hub/animal-experiments>

Comments for Authors to ensure the paper complies with the Statistics Policy (Required):

Please, make sure that all data point are included on the graph and are shown as mean +/- SD.

Comments to the Author (Required):

The manuscript has been reviewed by an expert in the field who felt the study was interesting and timely. In the revised version of the manuscript please address all the points raised by the reviewer.

Please also see 'Required Items' above.

Senior Editor:

Comments for Authors to ensure the paper complies with the Statistics Policy (Required):

Please provide exact p-values, unless $p < 0.001$.

Please show data points for individual experiments in all figures, unless $n > 30$.

Comments to the Author:

Thank you for submitting your manuscript to Journal of Physiology. It has been evaluated by an external referee and a reviewing editor, who were generally positive about the manuscript. The referee did however raise some significant concerns around caveats to your study and its interpretation that you should at least thoroughly address with a discussion in a revised version of your manuscript. All points that the referee raised should be addressed. Around presentation of statistics, I noted that you need to report exact p-values, except for $p < 0.001$, and also show data points for individual experiments in all figures (as also noted by the referee and reviewing editor). It does appear that you indicate SD values, as required, rather than SEMs.

REFEREE COMMENTS

Referee #1:

Manuscript Wu et al from a well-known Epilepsy/Astrocyte research group investigates the mechanisms of astrocytic cell death in KA model of TLE and whether and how preventing astrocytic death affects the epileptogenesis. Previously, authors reported that hippocampal astrocytes undergo rapid necroptosis, a form of cell death via activation of several inflammatory mediators including RIPK1, RIPK3, and MLKL. In the present study, the authors used several blockers of these mediators and observed a significant reduction in TLE-induced astrocytic loss. The authors further confirm these observations in the TNFR1 knockout condition. TNFR1 is a receptor for TNF α whose activation regulates the RIPK1 and other downstream mediators of necroptosis and can prevent astrocytic loss. These results show a promising rescue effect on the histopathological changes associated with TLE hippocampus including astrocytic density, reactive gliosis, granule cell dispersion, and CA1 subfield area. However, preventing these changes did not influence epileptogenesis as the authors did not see any changes in the seizure activity at acute as well as chronic stages.

Overall, this study elegantly demonstrates a causal association between the activation of inflammatory mediators and histopathological changes in the TLE. The data and presentation largely support the author's conclusions. However, it leaves readers with several burning questions that don't seem to be addressed adequately in the discussion. I have the following minor comments:

1. The reported histopathological changes are restricted to the hippocampus, however, seizures induced by KA can range from focal to generalized which can spread the pathology to other brain regions, especially the limbic structures. So, is it possible that other associated brain regions such as the amygdala and entorhinal cortex might have also undergone astrocytic necroptosis? If so, is it likely that the rescue strategies were not sufficient to generate a widespread effect in all brain regions showing astrocytic necroptosis? This may be the plausible reason why authors could not enfeeble the seizures/epileptogenesis despite significant rescue in the hippocampus.
2. Based on the above comment It may be unrealistic to expect changes in seizures at the whole brain or behavior levels upon rescuing astrocytic pathology in the hippocampus. I think the functional consequences of astrocytic rescue may be better reflected in ex-vivo hippocampal slices by recording the spontaneous field potential or hyperexcitability latency in the pyramidal neurons.
3. Do remaining astrocytes in the hippocampus show altered homeostatic functions? If yes, did rescuing the astrocytic density reinstate their homeostatic functions such as glu, GABA, and K $^{+}$ uptake thereby E-I balance? I guess, despite rescuing the numerical density of astrocytes, the homeostatic functions would have not been sufficiently normalized to reduce seizures.
4. Figures with the bar diagrams should have data points for consistency and transparency.
5. Fig 3D does not have significant labels.
6. In fig. 1B the label GFAP has unnecessary extra space masking the image.

END OF COMMENTS

EDITOR COMMENTS

Reviewing Editor:

Ethics Concerns:

Please make sure your article complies with our animal ethics policy:

<https://physoc.onlinelibrary.wiley.com/hub/animal-experiments>

Comments for Authors to ensure the paper complies with the Statistics Policy (Required):
Please, make sure that all data point are included on the graph and are shown as mean +/- SD.

Comments to the Author (Required):

The manuscript has been reviewed by an expert in the field who felt the study was interesting and timely. In the revised version of the manuscript please address all the points raised by the reviewer.

Please also see 'Required Items' above.

Senior Editor:

Comments for Authors to ensure the paper complies with the Statistics Policy (Required):
Please provide exact p-values, unless $p < 0.001$.

Please show data points for individual experiments in all figures, unless $n > 30$.

Comments to the Author:

Thank you for submitting your manuscript to Journal of Physiology. It has been evaluated by an external referee and a reviewing editor, who were generally positive about the manuscript. The referee did however raise some significant concerns around caveats to your study and its interpretation that you should at least thoroughly address with a discussion in a revised version of your manuscript. All points that the referee raised should be addressed. Around presentation of statistics, I noted that you need to report exact p-values, except for $p < 0.001$, and also show data points for individual experiments in all figures (as also noted by the referee and reviewing editor). It does appear that you indicate SD values, as required, rather than SEMs.

***Answer:** All reviewer comments have been addressed. The exact p-values and data points are specified in the revised manuscript. All data are displayed as mean \pm standard deviation (SD).*

REFEREE COMMENTS

Referee #1:

Manuscript Wu et al from a well-known Epilepsy/Astrocyte research group investigates the mechanisms of astrocytic cell death in KA model of TLE and whether and how preventing astrocytic death affects the epileptogenesis. Previously, authors reported that hippocampal astrocytes undergo rapid necroptosis, a form of cell death via activation of several inflammatory mediators including RIPK1, RIPK3, and MLKL. In the present study, the authors used several blockers of these mediators and observed a significant reduction in TLE-induced astrocytic loss. The authors further confirm these observations in the TNFR1 knockout condition. TNFR1 is a receptor for TNF α whose activation regulates the RIPK1 and other downstream mediators of necroptosis and can prevent astrocytic loss. These results show a promising rescue effect on the histopathological changes associated with TLE hippocampus including astrocytic density, reactive gliosis, granule cell dispersion, and CA1 subfield area. However, preventing these changes did not influence epileptogenesis as the authors did not see any changes in the seizure activity at acute as well as chronic stages.

Overall, this study elegantly demonstrates a causal association between the activation of inflammatory mediators and histopathological changes in the TLE. The data and presentation largely support the author's conclusions. However, it leaves readers with several burning questions that don't seem to be addressed adequately in the discussion. I have the following minor comments:

1. The reported histopathological changes are restricted to the hippocampus, however, seizures induced by KA can range from focal to generalized which can spread the pathology to other brain regions, especially the limbic structures. So, is it possible that other associated brain regions such as the amygdala and entorhinal cortex might have also undergone astrocytic necroptosis? If so, is it likely that the rescue strategies were not sufficient to generate a widespread effect in all brain regions showing astrocytic necroptosis? This may be the plausible reason why authors could not enfeeble the seizures/epileptogenesis despite significant rescue in the hippocampus.

Answer: We agree with the reviewer that it cannot be ruled out that astrocytes outside the ipsilateral hippocampus also undergo necroptotic death, especially since status epilepticus in our model is characterized by a series of secondary generalized seizures. On the other hand, we could not find any necroptotic death in the contralateral hippocampus, which argues against seizure-induced cell death outside the ipsilateral hippocampus. It should also be noted that Nec-1s was administered systemically (i.p.), so there is actually no plausible reason why it should inhibit necrosis only in the injected hippocampus and not in other brain regions. However, as we have not investigated this, we cannot rule it out. We have therefore discussed this point in the revised manuscript (page 16, line 14-18).

2. Based on the above comment It may be unrealistic to expect changes in seizures at the whole brain or behavior levels upon rescuing astrocytic pathology in the hippocampus. I think the functional consequences of astrocytic rescue may be better reflected in ex-vivo hippocampal slices by recording the spontaneous field potential or hyperexcitability latency in the pyramidal neurons.

Answer: As mentioned above, we do not believe that the protective effect of the i.p. injected Nec-1s was confined to the hippocampus, because Nec-1s was given systemically. Furthermore, in our model of focal epilepsy (TLE), the epileptogenic focus is most likely localized in the ipsilateral hippocampus. Therefore, we find it not at all unrealistic that the rescue of hippocampal pathologies can influence seizure activity. Investigating functional

consequences of astrocyte rescue in ex vivo hippocampal slices by recording spontaneous field potentials or hyperexcitability latency in pyramidal neurons is an excellent idea, which we will consider in our follow-up studies.

3. Do remaining astrocytes in the hippocampus show altered homeostatic functions? If yes, did rescuing the astrocytic density reinstate their homeostatic functions such as glu, GABA, and K⁺ uptake thereby E-I balance? I guess, despite rescuing the numerical density of astrocytes, the homeostatic functions would have not been sufficiently normalized to reduce seizures.

Answer: That is a very good point. In fact, the remaining astrocytes show altered homeostatic properties, as we demonstrated for instance in our 2015 Brain paper (Bedner et al., 2015). We totally agree that the rescue of astrocytes may not be sufficient to restore homeostatic functions and have therefore discussed this in more detail in the revised Discussion (page 16, line 11-14).

4. Figures with the bar diagrams should have data points for consistency and transparency.

Answer: The points have been added to the bar diagrams.

5. Fig 3D does not have significant labels.

Answer: Significant labels have been added to Fig. 3D.

6. In fig. 1B the label GFAP has unnecessary extra space masking the image.

Answer: The label has been changed accordingly.

Dear Professor Steinhäuser,

Re: JP-RP-2025-287565R1 "Targeting necroptosis protects against astrocyte death and hippocampal sclerosis in experimental temporal lobe epilepsy" by Zhou Wu, Lukas Henning, Christian Steinhäuser, and Peter Bedner

We are pleased to tell you that your paper has been accepted for publication in The Journal of Physiology.

Yours sincerely,

Nathan Schoppa
Senior Editor
The Journal of Physiology

If you would like to receive our 'Research Roundup', a monthly newsletter highlighting the cutting-edge research published in The Physiological Society's family of journals (The Journal of Physiology, Experimental Physiology, Physiological Reports, The Journal of Nutritional Physiology and The Journal of Precision Medicine: Health and Disease), please click this link, fill in your name and email address and select 'Research Roundup':
<https://www.physoc.org/journals-and-media/membernews>

- **TRANSPARENT PEER REVIEW POLICY:** To improve the transparency of its peer review process, The Journal of Physiology publishes online as supporting information the peer review history of all articles accepted for publication. Readers will have access to decision letters, including Editors' comments and referee reports, for each version of the manuscript as well as any author responses to peer review comments. Referees can decide whether or not they wish to be named on the peer review history document.
- You can help your research get the attention it deserves! Check out Wiley's free Promotion Guide for best-practice recommendations for promoting your work at: www.wileyauthors.com/eoo/guide. You can learn more about Wiley Editing Services which offers professional video, design, and writing services to create shareable video abstracts, infographics, conference posters, lay summaries, and research news stories for your research at: www.wileyauthors.com/eoo/promotion.
- **IMPORTANT NOTICE ABOUT OPEN ACCESS:** To assist authors whose funding agencies mandate public access to published research findings sooner than 12 months after publication, The Journal of Physiology allows authors to pay an Open Access (OA) fee to have their papers made freely available immediately on publication.

EDITOR COMMENTS

Reviewing Editor:

Many thanks for addressing the comments raised by the reviewer.

Senior Editor:

Congratulations! Your revised manuscript has been considered acceptable for publication.

REFeree COMMENTS

Referee #1:

Authors have addressed all of my concerns. I have no comments.